# PROPERTY-DRIVEN PROTEIN INVERSE FOLDING WITH MULTI-OBJECTIVE PREFERENCE ALIGNMENT

**Junqi Liu**[1,2,*]**, Xiaoyang Hou**[2,*]**, Chence Shi**[2,3,4]**, Xin Liu**[2]**, Zhi Yang**[1,†]**, Jian Tang**[3,5,6,2,†]

[1] School of Computer Science, Peking University [2] BioGeometry [3] Mila - Québec AI Institute
[4] Université de Montréal [5] HEC Montréal [6] CIFAR AI Research Chair

{liujunqi,yangzhi}@pku.edu.cn
houxiaoyangi7@gmail.com
jian.tang@hec.ca

## ABSTRACT

Protein sequence design must balance designability, defined as the ability to recover a target backbone, with multiple, often competing, developability properties such as solubility, thermostability, and expression. Existing approaches address these properties through post hoc mutation, inference-time biasing, or retraining on property-specific subsets, yet they are target dependent and demand substantial domain expertise or careful hyperparameter tuning. In this paper, we introduce ProtAlign, a multi-objective preference alignment framework that fine-tunes pretrained inverse folding models to satisfy diverse developability objectives while preserving structural fidelity. ProtAlign employs a semi-online Direct Preference Optimization strategy with a flexible preference margin to mitigate conflicts among competing objectives and constructs preference pairs using *in silico* property predictors. Applied to the widely used ProteinMPNN backbone, the resulting model MoMPNN enhances developability without compromising designability across tasks including sequence design for CATH 4.3 crystal structures, *de novo* generated backbones, and real-world binder design scenarios, making it an appealing framework for practical protein sequence design.

## 1    INTRODUCTION

Inverse folding is a fundamental task in protein design, spanning applications from refining sequences of natural proteins to generating sequences for *de novo* designed backbones (Yue & Dill, 1992; Notin et al., 2024; Khakzad et al., 2023). Substantial progress has been made with models trained to accurately recover sequences compatible with a target backbone, demonstrating strong capacity to capture structure–sequence relationships (Gao et al., 2023; Qiu et al., 2024b; Xue et al., 2025), and post-training approaches have been explored to further improve sequence quality (Widatalla et al., 2024; Xue et al., 2025; Xu et al., 2025; Wang et al., 2025). However, real-world design pipelines demand more than high sequence recovery: they typically require proteins that are both designable and developable, exhibiting properties such as solubility, thermostability, and expression level, with additional traits depending on specific design goals (Peterson et al., 2007; Salihu & Alam, 2015).

Several strategies have been explored to incorporate developability preferences into the generation process. **(1) Post-hoc mutation**: generate sequences with existing tools and then introduce mutations to improve properties. While simple, beneficial mutations are often sparse and difficult to identify (Broom et al., 2017). **(2) Inference-time biasing**: adjust amino acid sampling probabilities (Goverde et al., 2024) or use reward signals to guide sequence generation (Xiong et al., 2025). These techniques can introduce instability and require careful hyperparameter tuning to balance property optimization with sequence quality. **(3) Retraining on curated subsets**: construct datasets filtered for desired properties and retrain the model to implicitly learn the bias (Goverde et al., 2024;

---

[†] Corresponding authors.
[*] These authors contributed equally and share first authorship; each author may use an adjusted author order when appropriate. Work conducted during their internships at BioGeometry.

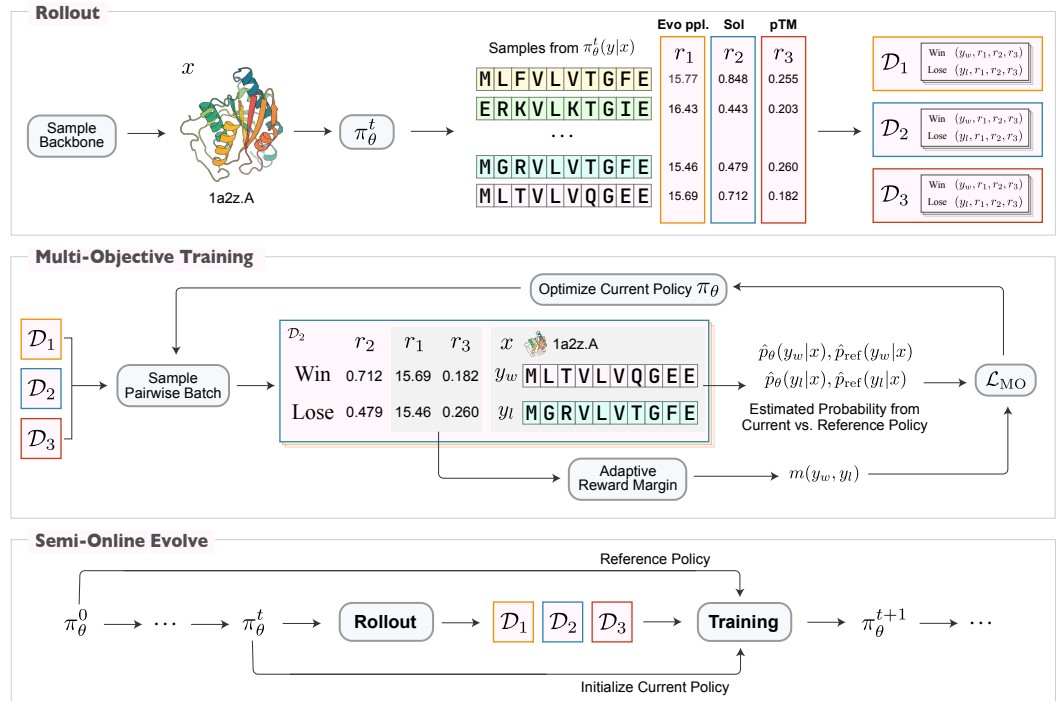

Figure 1: The ProtAlign framework. ProtAlign optimizes the policy model in a semi-online regime composed of alternating rollout and training stages. In the rollout stage, protein backbones are sampled from the training set, and the current policy model generates rollouts at a higher temperature. These rollouts are evaluated with property predictors, and pairwise preference datasets are constructed for each property. During training, pairwise entries are drawn evenly across the datasets, and an adaptive preference margin is introduced to resolve conflicts among multiple objectives.

Ertelt et al., 2024). Although such methods have achieved wet-lab validated success, they rely on carefully curated datasets and are difficult to generalize across diverse design objectives.

To address these challenges, we introduce ProtAlign, an optimization framework that aligns pretrained inverse folding models with both designability and diverse developability objectives. ProtAlign employs a novel semi-online Direct Preference Optimization algorithm with a flexible preference margin to balance competing goals (Figure 1). This strategy enables robust optimization for developability without sacrificing sequence–structure fidelity, even though developability metrics do not directly capture sequence-structure consistency. Training data are generated by annotating sequences with a suite of property predictors and forming pairwise preference sets for each property. During training, pairs are sampled evenly across properties, and the flexible margin in the DPO loss helps reconcile conflicting optimization directions. The overall training proceeds in a semi-online manner through iterative rollout, annotation, and updating, which avoids running property predictor models during training, thereby significantly reducing computational cost.

We instantiate our framework on ProteinMPNN (Dauparas et al., 2022), one of the most widely used inverse folding models, training it on the commonly adopted CATH dataset (Sillitoe et al., 2021) to obtain MoMPNN. We evaluate two key developability properties, i.e., solubility and thermostability, and show that MoMPNN outperforms subset-trained baselines such as SolubleMPNN (Goverde et al., 2024) and HyperMPNN (Ertelt et al., 2024), which are specifically designed for these properties. We comprehensively assess MoMPNN across redesigning sequences for crystal structures in the CATH 4.3 test set, designing sequences for *de novo* generated backbones, and applications to realistic binder design scenarios. MoMPNN shows superior performance across these diverse evaluation tasks, demonstrating the broad applicability and effectiveness of our alignment framework.

Our main contributions are:

- We propose a multi-objective alignment framework, ProtAlign, for optimizing protein inverse folding models towards arbitrary desired developability properties without compromising designability with semi-online multiple-objective preference optimization.

- Applying ProtAlign to ProteinMPNN, our resulting model MoMPNN achieves significant improvement on developability properties, outperforming existing baselines across crystal, *de novo* and real-scenario benchmarks.

- By adding *de novo* benchmarks and incorporating developability metrics into inverse folding evaluation, we offer a systematic framework for assessing model performance beyond recovery, thereby opening new avenues for future research.

## 2 RELATED WORK

**Protein Inverse Folding.**     Protein inverse folding is to generate a protein's amino acid sequences given its structure. Early work like GraphTrans (Carscadden et al., 2021), StructGNN (Chou et al., 2024) and Geometric Vector Perceptrons (GVPs) (Jing et al., 2020) utilize the graph neural network to design protein sequences. And, ProteinMPNN (Dauparas et al., 2022) extends GraphTrans by introducing more geometry features and random decoding. ESM-IF (Hsu et al., 2022) trains a large-scale inverse folding framework based. PiFold (Gao et al., 2022) accelerates sequence generation with a one-shot predicting strategy. FMIF (Nisonoff et al., 2024) explores applying flow matching to inverse folding. Additionally, LM-Design (Zheng et al., 2023), CarbonDesign (Ren et al., 2024) and InstructPLM (Qiu et al., 2024b) utilize protein language models in sequence design. There are also emerging approaches that move beyond traditional inverse folding (Song et al., 2024; Tang et al.; Wu et al.) to design functional sequences by considering not only the backbone structure but also the broader biochemical context of the protein. While ProteinMPNN is still the most widely used and wet-lab-verified model, we chose it as our backbone model during our evaluation.

**Preference Optimization.**     Recently, many methods to align LLMs with human feedback have emerged. To better align these models with human preferences, these methods can be categorized into two classes: online and offline methods. Online methods (Shao et al., 2024), such as PPO (Schulman et al., 2017), are typically employed to optimize policy models through direct reward optimization. While effective, online RL methods are computationally intensive and potentially unstable (Gupta et al., 2025). Recently, offline methods DPO (Rafailov et al., 2023) and its variants (Azar et al., 2024; Meng et al., 2024) have been introduced. They can align models to pairwise preferences rather than an explicit reward model directly. To address the possible overfitting and collapse problems of offline DPO (Guo et al., 2024a), several online or semi-online variants (Guo et al., 2024a; Calandriello et al., 2024; Lanchantin et al., 2025) have been developed. Inspired by these works, we design a semi-online approach combining the benefits of self-evolving and computational efficiency by separating the rollout phase from training.

Preference optimization has been used on inverse folding models to improve their performance. ProteinDPO (Widatalla et al., 2024) leverages an experimentally derived stability preference dataset to enhance the stability of designed sequences. ResiDPO (Xue et al., 2025) uses a residue-level labeled dataset to improve sequence designability. InstructPLM-DPO (Xu et al., 2025) is fine-tuned on a TM-Score-constructed dataset (Zhang & Skolnick, 2005) to better align sequence outputs with target structures. ProteinZero (Wang et al., 2025) directly defines a composite reward function combining TM-Score and energy, and optimizes through online GRPO. These works mainly focus on improving designability and cannot extend to developability properties that may conflict with designability.

**Multi-Objective Preference Alignment**     The multi-dimensional nature of human preferences (Vamplew et al., 2018; Bai et al., 2022) motivates various methods for handling multiple objectives (Wang et al., 2024). Early work explored parameter-merging approaches like rewarded soups (Jang et al., 2023; Lin et al., 2023; Rame et al., 2023; Wortsman et al., 2022), preference-conditioned prompting was introduced (Zhu et al., 2023; Basaklar et al., 2022; Guo et al., 2024b; Yang et al., 2024), enabling direct control over preference weightings (Zhou et al., 2023b). Beyond inference-time techniques, retraining-based strategies have emerged as a promising direction: MORLHF (Wu et al., 2023; Bai et al., 2022) incorporates multiple rewards through scalarization, and MODPO (Zhou et al., 2023a) integrates multi-objective optimization directly into reward learning, theoretically matching

MORLHF yet practically offering greater stability and efficiency. We leverage similar techniques when optimizing inverse folding models towards multiple objectives.

## 3 PRELIMINARIES

**ProteinMPNN.** ProteinMPNN (Dauparas et al., 2022) aims to model the conditional distribution $P(\text{seq} \mid x)$ and generate sequences that are compatible with the target structure given a backbone structure $x$. It is an order-agnostic autoregressive model that generates sequences conditioned on a given backbone structure. For a backbone $x$ with $L$ residues, the probability of a sequence $y = (y_1, \cdots, y_L)$ is factorized as:

$$\pi_\theta(y \mid x, \sigma) \;=\; \prod_{i=1}^{L} \pi_\theta(y_{\sigma(i)} \mid x, y_{\sigma(<i)}), \tag{1}$$

where $\sigma$ is a random permutation of residue indices that enforces order invariance.

The model is trained on structure–sequence pairs from the Protein Data Bank (PDB) using cross-entropy loss with teacher forcing. At each step, a random permutation $\sigma$ is sampled, and the loss is computed as

$$\mathcal{L}_{\text{CE}}(\theta) \;=\; -\mathbb{E}_{(x,y)\sim\mathcal{D}} \; \mathbb{E}_\sigma \left[ \sum_{i=1}^{L} \log \pi_\theta\big(y_{\sigma(i)} \mid x, y_{\sigma(<i)}\big) \right].$$

**Direct Preference Optimization.** Direct Preference Optimization (DPO) (Rafailov et al., 2023) is a recent framework for aligning generative models with human or task-specific preferences without requiring explicit reward modeling. The key idea is to learn from pairwise preference data: given a context $x$ (e.g., a protein backbone) and two candidate outputs $y_w$ (preferred) and $y_l$ (less preferred), the preference model assumes

$$P(y_w \succ y_l \mid x) = f(r(y_w), r(y_l)),$$

where $r(\cdot)$ is an implicit reward function and $f$ is typically modeled as a logistic function over reward differences. Instead of explicitly estimating $r$ from pairwise data, DPO derives a tractable training loss by enforcing that the conditional likelihood ratio between the fine-tuned policy $\pi_\theta$ and the reference policy $\pi_{\text{ref}}$ matches the observed preference:

$$\mathcal{L}_{\text{DPO}}(\theta) = -\mathbb{E}_{(x,y_w,y_l)} \left[ \log \sigma\big(\beta \log \tfrac{\pi_\theta(y_w|x)}{\pi_{\text{ref}}(y_w|x)} - \beta \log \tfrac{\pi_\theta(y_l|x)}{\pi_{\text{ref}}(y_l|x)}\big) \right],$$

where $\sigma$ is the sigmoid function and $\beta$ controls the strength of preference alignment. This objective directly optimizes the model to prefer $y_w$ over $y_l$, while regularizing towards the reference model.

## 4 METHOD

In this section, we present our multi-objective preference alignment framework ProtAlign. Firstly, we describe our method of multi-objective optimization in protein sequence design tasks in Section 4.2. Then, we outline our semi-online multi-objective training strategy for efficient exploration in Section 4.3. Finally, Section 4.4 introduces the construction of preference pairs.

### 4.1 NOTATIONS

Let $x$ denote the input backbone structure, $y$ a protein sequence, and $\theta$ the model parameters. We write $\pi_\theta$ for the sequence conditional distribution induced by $\theta$, and $\pi_{\text{ref}}$ for the reference model obtained prior to post-training. A superscript $t$ indicates variables in the $t$-th stage of semi-online training, such as $\pi^t$ or $y^t$. Let $K$ be the number of properties, and let $\{M_K : (x,y) \to \mathbb{R}\}$ denote the corresponding *in silico* predictors. Finally, let $\mathcal{D} = [\mathcal{D}_1, \cdots, \mathcal{D}_K]$ represent the pairwise datasets for each property, generated by the respective $M_k$.

## 4.2 Multi-Objective Optimization with Flexible Preference Margin

Our optimization process aligns a pretrained inverse folding model based on a series of pairwise datasets $\{\mathcal{D}_k\}$ for each target property $k$, annotated by in silico predictors $M_k$ (Section 4.4). The ultimate goal is to simultaneously improve the model's performance on all properties while maintaining limited divergence from the original model. Formally, we maximize the following objective:

$$\arg\max_{\theta} \mathcal{L}(\pi_\theta) = \mathbb{E}_{x \sim \mathcal{D}_x, y \sim \pi(\cdot|x)} \left[ \sum_k w_k r_k(x, y) \right] - \beta \mathbb{D}_{\text{KL}}(\pi_\theta(y|x) \| \pi_{\text{ref}}(y|x)), \qquad (2)$$

where $\mathcal{D}_x$ is the distribution of possible protein backbones, $r_k$ is the implicit reward function from $\mathcal{D}_k$ and $w_k$ is an adjustable weight.

As in the original Direct Preference Optimization (DPO) derivation, we assume the preference relations follow the Bradley-Terry model:

$$p^*(y_1 \succ y_2 \mid x) = \frac{\exp(r(x, y_1))}{\exp(r(x, y_1)) + \exp(r(x, y_2))} = \sigma\big(r(x, y_1) - r(x, y_2)\big). \qquad (3)$$

We integrate the multi-property policy objective in Eq. 2 with the pairwise preference model in Eq. 3 (details in the Appendix B.1) (Zhou et al., 2023a), we derive a flexible-margin Direct Preference Optimization (DPO) loss, denoted $\mathcal{L}_{\text{MO}}(\theta; \mathbf{D}_k)$. It explicitly accounts for both multi-property rewards and adaptive preference margins. The intuition of adaptive preference margins is that if $y_w$ performs worse than $y_l$ on some auxiliary property, the required margin for this pair should be reduced, preventing conflicting optimization from overemphasizing a single property at the cost of others.

$$\mathcal{L}_{\text{MO}}(\theta; \mathcal{D}_k) = -\mathbb{E}_{(x, y_w, y_l) \sim \mathcal{D}_k} \left[ \log \sigma \Big( w_k \big( \beta \log \frac{\pi_\theta(y_w|x)}{\pi_{\text{ref}}(y_w|x)} - \beta \log \frac{\pi_\theta(y_l|x)}{\pi_{\text{ref}}(y_l|x)} - m_k(y_w, y_l) \big) \Big) \right],$$

$$m_k(y_w, y_l) = \lambda \sum_{k' \neq k} w_r \big( r_{k'}(x, y_w) - r_{k'}(x, y_l) \big). \qquad (4)$$

During training, we sample entries from the pairwise datasets $\mathcal{D}_k$ evenly. The adaptive margin $m(y_w, y_l)$ is precomputed before training with our property predictors.

Next, we describe the definition of ProteinMPNN's probability term $\pi_\theta$ in the loss function $\mathcal{L}_{\text{MO}}$. Unlike LLMs, ProteinMPNN is not a left-to-right model but an order-agnostic autoregressive model. While $\pi_\theta$ can be easily calculated for left-to-right causal models, its exact estimation for order-agnostic models inherently requires extensive sampling across different decoding orders. We adopt an efficient approach for estimating the log-ratio in our loss function inspired by recent works on discrete diffusion-based LLMs (Zhu et al., 2025). The probabilities $\pi_\theta$ and $\pi_{\text{ref}}$ of an order-agnostic inverse folding model by sampling multiple random residue orders. Crucially, both models are evaluated under the *same sampled orders*:

$$\hat{p}_\theta(y \mid x) = \frac{1}{K} \sum_{k=1}^{K} \pi_\theta(y \mid x, \sigma_k), \quad \hat{p}_{\text{ref}}(y \mid x) = \frac{1}{K} \sum_{k=1}^{K} \pi_{\text{ref}}(y \mid x, \sigma_k), \qquad (5)$$

where $\{\sigma_k\}_{k=1}^{K}$ are the same sampled permutations. This shared-order evaluation significantly reduces the variance of the estimated log-ratio, leading to more stable optimization.

## 4.3 Semi-Online Training for Efficient Exploration

It is widely acknowledged that online exploration plays an important role in RL alignment (Tang et al., 2024). However, such training regime requires significant resource and infrastructure engineering for rollout and evaluation during training. Fortunately, semi-online training has been shown to

be as effective as pure online training (Lanchantin et al., 2025). Thus, we build a semi-online DPO framework to decouple rollout and evaluation from training, which allows for efficient batch computation and is easy to deploy.

As detailed in Algorithm 1, the semi-online DPO framework proceeds in an iterative manner. At each iteration $t$, the current policy $\pi_\theta^t$ first generates rollout sequences under a rollout temperature $\tau$, which is deliberately set higher than the evaluation temperature in order to promote diversity. These rollouts are subsequently evaluated by $K$ property predictors, from which pairwise preference datasets $\{\mathcal{D}_k\}$ are constructed. The model is then optimized on the newly generated preference data for several steps, yielding the updated policy $\pi_\theta^{t+1}$. Overall, this paradigm integrates the advantages of both online and offline learning: the model alternates between online data generation and update across iterations, while the optimization within each iteration is performed in an offline mode. In addition, our approach requires no modification to the property predictors, ensuring strong compatibility with existing methods; it allows batch inference to maximize resource utilization; and each predictor can fully exploit the available computational capacity without introducing additional overhead.

---

**Algorithm 1** Iterative Training Algorithm for Semi-Online DPO

---

1: **Input:** Base model $\pi_0$, Preference predictors $\{M_1, M_2, \ldots, M_K\}$, Preference weights $\{w_1, w_2, \ldots, w_K\}$, Backbone dataset $\mathcal{X}$, Number of iterations $T$, Number of designed sequences per backbone $n$, Sampling temperature $\tau$, Number of sampling backbones at each iteration $N$.
2: Initialize model $\pi_\theta = \pi_0$.
3: **for** $t = 1$ to $T$ **do**
4:     **for** $i = 1$ to $N$ **do**
5:         Sample backbone $x \leftarrow \text{Sample}(\mathcal{X})$ .
6:         Generate $n$ sequences per backbone $S = \{s_i\}_{i=1}^n \sim \pi_\theta(\cdot \mid x, \tau)$.
7:         Use reward models $\{M_1, M_2, ..., M_K\}$ to calculate rewards for each sample.
8:         **for** $k = 1$ to $K$ **do**
9:             Construct preference pairs for each reward $D_k^t = \{(y_w, y_l)\}$.
10:         **end for**
11:     **end for**
12:     Update model parameters $\theta^t \leftarrow \theta^{t-1} - \alpha \nabla_\theta \left( \sum_{k=1}^K w_k \mathcal{L}_{\text{MO}}(\theta; \mathcal{D}_k) \right)$.
13: **end for**
14: **Output:** Final optimized model $\pi_\theta$

---

## 4.4 Construction of Preference Datasets

We leverage existing protein property predictors as proxy annotators to provide pairwise preferences, building separate datasets for each property $k$. Given a backbone with $N$ candidate sequences, each sequence is scored with $M_k(y)$ and ranked accordingly. Following Xu et al. (2025), the $i$-th ranked sequence is paired with the $(N/2 + i)$-th ranked sequence ($i \leq N/2$), denoted $y_w$ and $y_l$. A pair $(y_w, y_l)$ is included in the dataset $\mathcal{D}_k$ only if the score gap satisfies $M_k(y_w) - M_k(y_l) > \delta_k$, where $\delta_k$ is a property-specific threshold. This procedure filters out ambiguous comparisons and yields consistent annotations from which DPO can learn implicit reward signals.

To capture diverse aspects of protein design, we categorize properties into two classes. **Designability properties** measure structural consistency between designed sequences and the input backbone, such as TM-score between a predicted structure and the target backbone or confidence metrics reported by structure prediction models. These metrics reflect the fundamental ability of an inverse folding model. **Developability properties**, in contrast, do not directly compare the designed sequence to the backbone and are primarily concerned with whether the protein sequence can achieve the intended purpose. We consider two main types: (1) General quality metrics, assessed for example by pseudo-likelihood scores from protein language models such as ESM (Lin et al., 2022), which correlate with evolutionary plausibility and often predict downstream outcomes such as solubility or expression (Adaptyv Bio, 2024); and (2) Targeted quality metrics, which capture properties directly related to whether the designed sequence can fulfill desired purposes, such as solubility and thermostability. These are important for practical use and are typically approximated by *in silico* predictors given the expense of wet-lab assays. Since developability properties does not consider the consistency between sequence and input structure, we jointly optimize developability properties

together with designability properties in our multi-objective alignment framework to ensure structural consistency while optimizing for desired developability.

# 5 EXPERIMENTS

We evaluate our framework by fine-tuning the widely adopted ProteinMPNN model toward two critical functional properties, solubility and thermostability, thereby deriving the MoMPNN models. Section 5.1 presents the details of model training and our evaluation setup. In Section 5.2, we assess MoMPNN's ability to redesign sequences for crystal structures in the CATH4.3 test set. Section 5.3 extends the evaluation to a more practical setting on *de novo* generated protein backbones. Finally, Section 5.4 focuses on a more specific application, evaluating MoMPNN's performance in designing sequences for *de novo* binders against a set of challenging protein targets.

## 5.1 EXPERIMENTAL SETUP

**Training and Testing.** Our model is trained on the CATH 4.3 training set (Orengo et al., 1997) based on the train-test-validation split referenced in (Hsu et al., 2022). During training, we generate eight sequences at a temperature of 1.0 for each sampled structure to encourage diversity. We use a temperature of 0.1 during evaluation for ProteinMPNN-related models, while other baselines are evaluated at their recommended temperature. More details on the preparation of the training and testing datasets are provided in the Appendix C.

**Baseline Methods.** We compare our model against representative methods from three categories: (1) state-of-the-art inverse folding models, including ProteinMPNN (Dauparas et al., 2022), ESM-IF (Hsu et al., 2022), and InstructPLM (Qiu et al., 2024a); (2) RL-based DPO method ProteinDPO (Widatalla et al., 2024); (3) task-specific models trained on protein subsets, SolubleMPNN (Goverde et al., 2024) for solubility and HyperMPNN (Ertelt et al., 2024) for thermostability; and (4) guidance-based methods, where we use SolubleMPNN and HyperMPNN as conditional models to guide the original ProteinMPNN model, following the approach of Nisonoff et al. (2024), resulting in Guidance[Sol] and Guidance[Thermo]. Additional details are provided in the Appendix B.3.

**Property Predictors.** We employ several computational predictors as in silico proxies for protein properties. For designability, we use the TM-score (TM), computed between ESMFold-predicted (Lin et al., 2022) structures and reference structures, or alternatively the pTM score from AlphaFold2 (Jumper et al., 2021) using the Initial Guess (IG) (Bennett et al., 2023) approach. For developability, we adopt Protein-Sol (Hebditch et al., 2017) as a widely used proxy for solubility (Sol), TemBERTure (Rodella et al., 2024), a model trained on large-scale datasets, as a proxy for thermostability (Thermo), and the pseudo-likelihood score from the ESM-2 model (Lin et al., 2022) (Evolutionary Perplexity, EP) as an indicator of sequence quality. In our experiments, we systematically compare the benefits of training with different combinations of these properties. Additional details are provided in the Appendix B.4.

## 5.2 SEQUENCE REDESIGN FOR CATH4.3 CRYSTAL STRUCTURES

We first evaluate our model on the CATH4.3 test set, a benchmark dataset commonly used for protein inverse folding models. CATH4.3 is a classification dataset of observed crystal structures, and performing inverse folding on these protein backbones reflects a model's ability to redesign sequences based on experimentally determined crystal structures. It is also worth noting that some ground-truth sequences in CATH are not soluble, and most of them are not thermally stable, so the solubility and thermostability of ground-truth sequences are low.

As shown in Table 1, our MoMPNN preserves the designability level of ProteinMPNN while significantly enhancing developability, achieving the best solubility and thermostability by explicitly optimizing for multiple desired properties. ProteinMPNN and other inverse folding baselines show high designability and moderate developability. SolubleMPNN achieves strong solubility and maintains reasonable structural quality. HyperMPNN could reach high thermostability but suffers from degraded designability, likely due to its smaller training data. Guidance-based methods partially address the trade-off by striking a balance between the property gains of subset-trained models and the

---

[1]InstructPLM was trained on the 4.2 version of the CATH dataset.

designability of ProteinMPNN. We also find that higher amino acid recovery does not correlate with higher designability and developability, and in a practical perspective, we focus more on designability and developability rather than the amino acid recovery.

We further analyze the results to understand how different objectives shape model behavior. TM leads to higher TM-scores and thus slightly stronger structural consistency, while IG consistently yields lower evolutionary perplexity, since it evaluates not only whether a sequence can refold but also how confident AlphaFold is in that prediction. In addition, directly incorporating EP does not substantially improve evolutionary plausibility, but it consistently enhances non-targeted metrics, serving as a useful regularizer that complements both TM and IG. To assess whether MoMPNN effectively captures the underlying patterns of protein solubility and thermal stability, we conducted an in-depth statistical analysis of generation sequences in Appendix A.1. We also provide the early results comparing multi-objective optimizing strategies in Appendix A.2.

## 5.3 SEQUENCE DESIGN FOR *de novo* GENERATED BACKBONES

We next extend our evaluation to a setting that more closely reflects practical protein design workflows, where we generate sequences for *de novo* backbones produced by RFDiffusion. We designed 4 unconditional backbones for each length in the range of [50, 500] with RFDiffusion as the input of the inverse folding models. This represents a more common application scenario for inverse folding models, serving as a tool to identify suitable sequences for newly designed protein backbones. Details of this *de novo* benchmark set are provided in Appendix C.2.

As shown in Table 2, MoMPNN demonstrates the strongest overall performance in this setting, even surpassing ProteinMPNN in structural consistency. ESM-IF and InstructPLM exhibit a substantial performance drop under *de novo* conditions, consistent with previous reports (Ren et al., 2024), whereas ProteinMPNN retains performance levels similar to those observed on crystal structures. SolubleMPNN achieves markedly better structural consistency than ProteinMPNN, but our models consistently outperform this baseline. In terms of training objectives, we observe that IG-based optimization yields higher structural consistency than TM in the *de novo* setting, while other phenomena remain consistent with the observations from CATH4.3.

Table 1: Comparison of protein sequence design methods on the CATH 4.3 test set across various metrics. Results for our RL-based MoMPNN trained with different annotator combinations (TM, IG, EP, Sol, and Thermo) are shown. The best and second-best values are highlighted in **bold** and underlined, respectively.

| Method | Designability Metrics | | | Developability Metrics | | | AAR ↑ |
|---|---|---|---|---|---|---|---|
| | RMSD ↓ | TM score ↑ | PLDDT ↑ | EP ↓ | Sol ↑ | Thermo ↑ | |
| Test Dataset | 3.97 | 0.761 | 80.8 | 5.80 | 0.620 | 0.246 | 1.000 |
| ESM-IF | 4.36 | 0.737 | 78.4 | 6.11 | 0.733 | 0.719 | 0.464 |
| InstructPLM [1] (default) | 6.81 | 0.628 | 73.4 | 7.97 | 0.653 | 0.396 | **0.574** |
| InstructPLM (T=0.1) | 6.96 | 0.632 | 74.4 | 7.31 | 0.657 | 0.455 | 0.584 |
| ProteinMPNN | **4.30** | 0.740 | 79.1 | 6.70 | 0.719 | 0.769 | 0.389 |
| ProteinDPO | 5.49 | 0.667 | 72.0 | 10.50 | 0.629 | 0.357 | 0.388 |
| SolubleMPNN | 4.48 | 0.733 | 78.8 | 6.54 | 0.794 | 0.815 | 0.382 |
| Guidance [Sol] | 4.33 | 0.740 | 79.4 | 6.40 | 0.762 | 0.805 | 0.393 |
| MoMPNN [Sol+TM] | 4.37 | 0.738 | 79.3 | 6.27 | **0.884** | 0.747 | 0.384 |
| MoMPNN [Sol+TM+EP] | 4.38 | 0.739 | **79.5** | 6.18 | 0.852 | 0.790 | 0.387 |
| MoMPNN [Sol+IG] | 4.73 | 0.727 | 79.3 | 6.00 | 0.883 | 0.751 | 0.382 |
| MoMPNN [Sol+IG+EP] | 4.61 | 0.731 | 79.3 | 5.99 | 0.856 | 0.789 | 0.384 |
| HyperMPNN | 4.90 | 0.706 | 74.3 | 7.81 | 0.719 | 0.929 | 0.359 |
| Guidance [Thermo] | **4.30** | 0.737 | 77.6 | 6.88 | 0.735 | 0.901 | 0.386 |
| MoMPNN [Thermo+TM] | **4.30** | 0.739 | 78.4 | 6.24 | 0.704 | 0.947 | 0.386 |
| MoMPNN [Thermo+TM+EP] | **4.30** | **0.742** | 78.6 | 6.12 | 0.731 | 0.946 | 0.387 |
| MoMPNN [Thermo+IG] | 4.38 | 0.734 | 78.2 | **5.85** | 0.694 | **0.963** | 0.382 |
| MoMPNN [Thermo+IG+EP] | 4.37 | 0.737 | 78.5 | 5.97 | 0.723 | 0.947 | 0.385 |

Table 2: Comparison of protein sequence design methods across different evaluation metrics on *de novo* backbone structures from RFDiffusion. The best results and the second-best results are marked **bold** and underlined. Note that AAR is not evaluated in this setting, as alignment to reference sequences is not applicable in *de novo* design.

| Method | Designability Metrics | | | Developability Metrics | | |
|---|---|---|---|---|---|---|
| | RMSD ↓ | TM score ↑ | PLDDT ↑ | EP ↓ | Sol ↑ | Thermo ↑ |
| ESM-IF | 13.51 | 0.461 | 57.6 | 7.27 | 0.616 | 0.806 |
| InstructPLM (default) | 22.44 | 0.134 | 32.6 | **3.33** | 0.539 | 0.278 |
| InstructPLM (T=0.1) | 22.58 | 0.132 | 33.5 | **2.73** | 0.538 | 0.367 |
| ProteinMPNN | 6.86 | 0.718 | 70.0 | 8.32 | 0.731 | 0.978 |
| ProteinDPO | 16.77 | 0.296 | 43.5 | 14.70 | 0.596 | 0.145 |
| SolubleMPNN | 6.61 | 0.733 | 70.5 | 8.36 | 0.799 | 0.992 |
| Guidance [Sol] | 6.20 | 0.748 | 71.8 | 8.14 | 0.774 | 0.989 |
| MoMPNN [Sol+TM] | 6.59 | 0.734 | 70.9 | 7.42 | **0.869** | 0.987 |
| MoMPNN [Sol+TM+EP] | 6.40 | 0.742 | 71.3 | 7.47 | 0.843 | 0.993 |
| MoMPNN [Sol+IG] | 6.37 | 0.742 | 71.5 | 7.21 | 0.867 | 0.983 |
| MoMPNN [Sol+IG+EP] | 6.17 | **0.751** | **72.0** | 7.34 | 0.843 | 0.993 |
| HyperMPNN | 7.51 | 0.693 | 68.0 | 8.25 | 0.727 | 0.992 |
| Guidance [Thermo] | 6.34 | 0.743 | 71.5 | 7.88 | 0.757 | 0.993 |
| MoMPNN [Thermo+TM] | 6.29 | 0.744 | 70.8 | 7.75 | 0.704 | 0.997 |
| MoMPNN [Thermo+TM+EP] | 6.48 | 0.737 | 70.5 | 7.64 | 0.736 | **0.999** |
| MoMPNN [Thermo+IG] | **6.14** | 0.748 | 71.1 | 7.32 | 0.684 | 0.998 |
| MoMPNN [Thermo+IG+EP] | 6.20 | 0.748 | 71.2 | 7.44 | 0.723 | 0.998 |

## 5.4 SEQUENCE DESIGN FOR *de novo* BINDERS

In this section, we extend our evaluation to specific design tasks, in which inverse folding models are used to design sequences for *de novo* binders generated by RFDiffusion. These binders target a set of challenging proteins, providing a practical assessment of the model's potential to support real-world applications. The inverse folding models gets 100 de novo backbones are designed for each binder as inputs and are required to generate 8 sequences for each backbone. A sequence is considered success if: binder sequence pLDDT > 80, inter-chain PAE < 10, and overall $C\alpha$ RMSD <

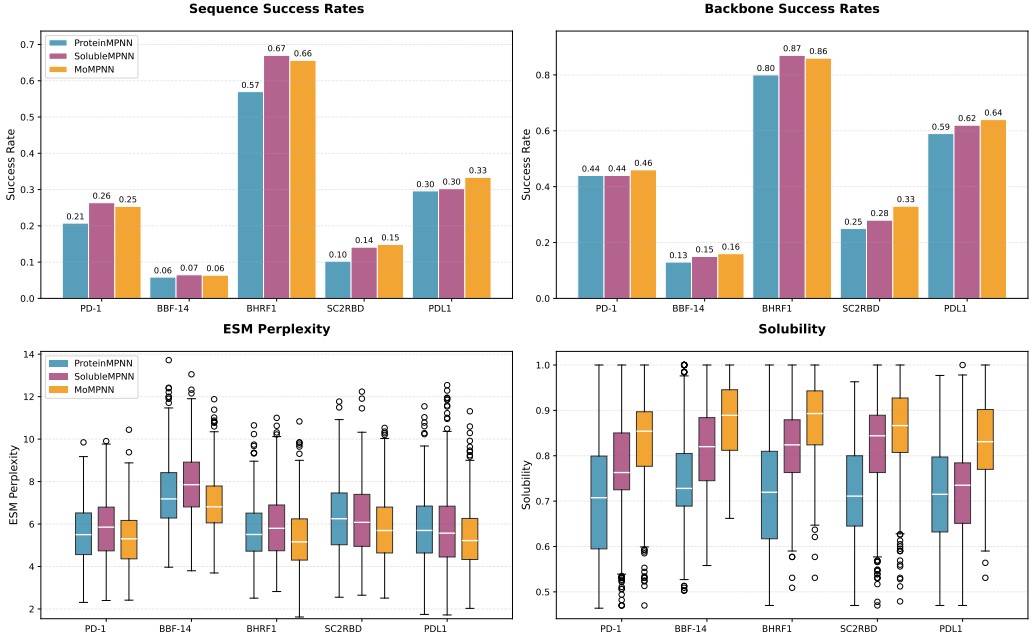

Figure 2: The result for ProteinMPNN, SolubleMPNN and MoMPNN on the binder design benchmark.

2 Å. A backbone is considered success if the model generates at least one success sequence for it. Details of the evaluation pipeline are provided in Appendix C.3.

As illustrated in Figure 2, our soluble variant MoMPNN [Sol+IG+EP] exhibits slightly higher success rates in both sequence and backbone than ProteinMPNN. It also achieves substantial performance gains over ProteinMPNN across two developability properties: evolutionary plausibility and solubility. Besides, MoMPNN performs on par with SolubleMPNN on designability, with better evolutionary plausibility and solubility. These results collectively demonstrate that MoMPNN preserves the essential capacity for binder design despite being post-trained only on monomeric inputs, and that the improvements in developability translate into complex settings without sacrificing designability.

## 6 CONCLUSION

We presented ProtAlign, a multi-objective alignment framework that extends inverse folding models beyond sequence recovery to jointly optimize for designability and diverse developability properties. By introducing a semi-online Direct Preference Optimization algorithm with a flexible preference margin, our approach achieves robust improvements in solubility and thermostability without compromising sequence–structure fidelity. Applied to ProteinMPNN, the resulting MoMPNN consistently outperforms subset-trained baselines across crystal, *de novo*, and real-world design tasks, highlighting the effectiveness and generality of our framework for practical protein engineering.

The limitations of MoMPNN primarily include the following two aspects. First, while our experiments have validated the model's effectiveness through various metrics, wet-lab experimental verification is still lacking. Second, this study primarily focuses on protein monomer properties; although testing was conducted on binders, no exploration was performed on complex-specific properties, which we will further investigate in subsequent work.

## ACKNOWLEDGMENT

We thank Tian Zhu for helpful discussions and suggestions, and Martin Pacesa for providing the BindCraft target files.

## REPRODUCIBILITY STATEMENT

We provide a detailed description of our algorithm in the Method section, ensuring that all steps of the approach are clearly explained. The hyperparameters used in our experiments are listed in Appendix B.2. We will release the source code and checkpoints in `https://github.com/Qivon7/MoMPNN`.

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

APPENDIX

# A ADDITIONAL EXPERIMENTS RESULTS

## A.1 IN-DEPTH ANALYSIS OF THE GENERATED SEQUENCES

**Solubility.** In order to systematically evaluate the solubility-related physical properties of the generated proteins, we calculated a series of quantitative indicators. These descriptors reflect different aspects of amino acid composition, surface exposure, and charge distribution, which together provide a comprehensive view of protein solubility and stability. The indicators include:

- **Overall Hydrophilic Residue Fraction**: the proportion of hydrophilic residues across the whole protein. A higher value indicates greater overall hydrophilicity and thus better solubility.

- **Surface Hydrophilic Residue Fraction** and **Surface Strong Hydrophilic Residue Fraction**: the fraction of hydrophilic residues (or strongly hydrophilic residues such as charged side chains) exposed on the protein surface. Higher fractions suggest stronger potential for hydrogen bonding or electrostatic interactions with water molecules.

- **Surface Hydrophilic SASA Fraction**: the proportion of solvent-accessible surface area (SASA) contributed by hydrophilic residues, directly reflecting whether these residues are exposed to solvent instead of buried within the protein core.

- **Surface Net Charge per 100 Residues**: the normalized net surface charge. Values farther from zero indicate stronger net charges, which promote electrostatic repulsion between protein molecules and reduce aggregation.

- **Surface Charge Distribution Uniformity**: a measure of how evenly charges are distributed across the protein surface. Higher uniformity implies a more balanced and ordered charge pattern, favoring stability in solution.

- **GRAVY Value (Grand Average of Hydropathy)**: an overall measure of hydropathy. Lower GRAVY values correspond to higher hydrophilicity, typically associated with enhanced solubility.

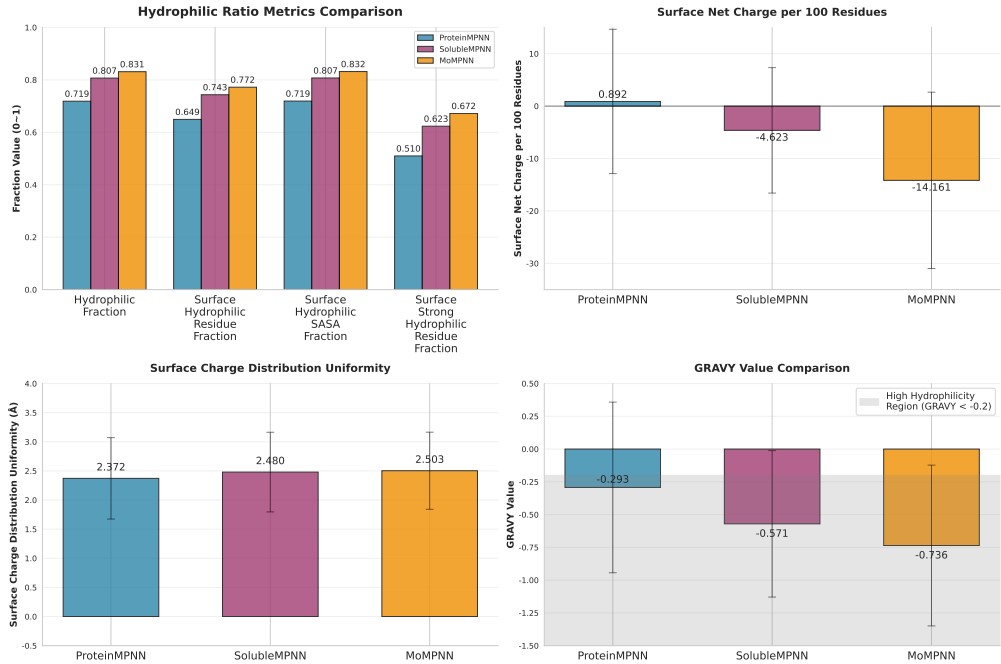

Figure 3: Quantative Analysis of ProteinMPNN, SolubleMPNN and MoMPNN generated sequences on hydrophilic-related metrics.

Results of MoMPNN [Sol+IG+ESM], ProteinMPNN, and SolubleMPNN based on sequences generated in the CATH benchmark are shown in Figure 3. Across all these indicators, MoMPNN consistently outperformed SolubleMPNN, suggesting that the proteins generated by MoMPNN not only achieves high in silico score but also exhibit more favorable distributions of surface charge and hydrophilic residues. This highlights the ability of MoMPNN to design proteins with genuinely improved solubility.

**Thermostability.** Since thermostability is a more challenging property to evaluate purely from an *in silico* perspective, we followed the analysis strategy of Ertelt et al. (2024) and examined the amino acid distribution of MoMPNN [Thermo+IG+ESM], HyperMPNN, and ProteinMPNN in both surface and core regions. The results based on sequences generated in the CATH benchmark are shown in Figure 4. MoMPNN and HyperMPNN display almost identical redistribution patterns across residue categories, which differ systematically from ProteinMPNN.

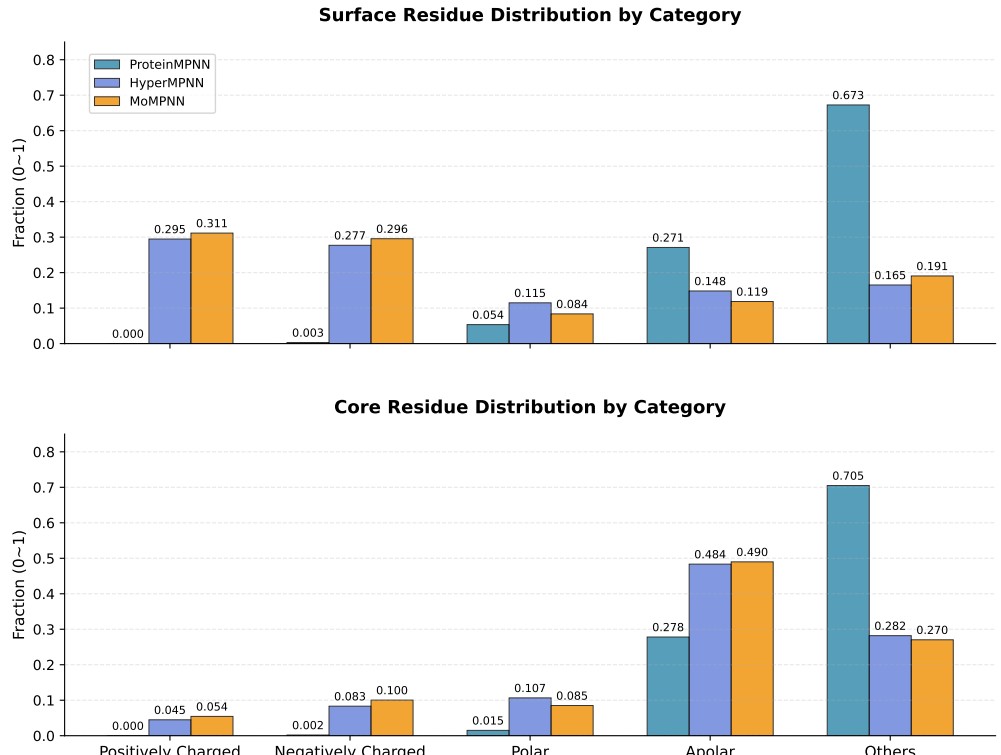

Figure 4: Differences in amino acid composition of proteins from ProteinMPNN, HyperMPNN and MoMPNN.

- **Positively charged residues (Lys, Arg).** Both MoMPNN and HyperMPNN show a clear increase on the surface and in the core compared with ProteinMPNN. This enrichment of positively charged residues strengthens electrostatic interactions: on the surface it enhances solubility, while in the core it facilitates stabilizing salt-bridge formation, contributing to thermostability.

- **Negatively charged residues (Asp, Glu).** Slightly increased in both surface and core for MoMPNN and HyperMPNN relative to ProteinMPNN. This modest enrichment enhances polarity at the surface and contributes to a more balanced charge distribution, which helps stabilize the folded structure.

- **Polar residues (Asn, Gln, Ser, Thr).** Both models show a small increase on surface and in core compared to ProteinMPNN. These residues can form hydrogen bonds that stabilize secondary structures and packing, providing moderate contributions to thermostability.

- **Apolar residues (Iso, Leu, Met, Phe, Trp, Tyr, Val).** In both MoMPNN and HyperMPNN, apolar residues are markedly decreased on the surface but strongly increased in the core.

> This redistribution is favorable: reduced hydrophobic exposure on the surface prevents aggregation, while enriched apolar residues in the core reinforce hydrophobic packing, a critical factor for thermostability.
>
> • **Other residues (Ala, Cys, Gly, His, Pro).** Both MoMPNN and HyperMPNN show a clear overall decrease. As residues in this category often introduce backbone flexibility, their reduction suggests a preference for more rigid and stable structural configurations.

Overall, the highly consistent residue redistribution patterns observed in MoMPNN and HyperMPNN, as compared with ProteinMPNN, indicate that MoMPNN inherits the stability-oriented features of HyperMPNN while maintaining a favorable balance between surface polarity and core hydrophobicity. These trends strongly support the ability of MoMPNN to design sequences with enhanced thermostability.

## A.2 PRELIMINARY EXPERIMENT RESULTS

In the early stage of our development, we conducted a small scale test to verify whether the current choice of multi-objective modeling leads to better performance, comparing it to a naive weighted score method introduced in B.3. CATH4.3 was employed as the training set, which is the same as our main experiment. Evaluation results are calculated by generating 16 sequences for each backbone in a curated validation set of 100 structures. All models use the default temperature for sampling as described in the main text.

In this experiment, we choose the Inital Guess and Evo. ppl as the optimization objectives. We also designed a baseline method, Weighted-score DPO, which aggregates multiple optimization objectives into a single score using weights and then performs optimization following the standard single-objective DPO framework. The weights used here are consistent with those of MoMPNN [IG+ESM].

According to Table 3, Weighted-score DPO achieves the best performance in terms of Evo ppl., but its performance on other metrics is inferior to that of the base model ProteinMPNN. MoMPNN achieve significantly more balanced results in the small-scale test set.

Table 3: Comparison of protein sequence design methods across different evaluation metrics on *de novo* backbone structures from RFDiffusion. The best results and the second-best results are marked **bold** and bold.

| Method | Designability Metrics | | | Evo. ppl ↓ | AAR ↑ |
|---|---|---|---|---|---|
| | RMSD ↓ | TM score ↑ | PLDDT ↑ | | |
| ESM-IF | 4.033 | 0.805 | 80.55 | 6.538 | 0.509 |
| InstructPLM | 7.196 | 0.683 | 74.77 | 6.946 | 0.607 |
| ProteinMPNN | 3.658 | 0.823 | 82.09 | 6.843 | 0.441 |
| Weighted-score DPO | 3.783 | 0.811 | 80.74 | 5.843 | 0.410 |
| MoMPNN [IG+ESM] | 3.706 | 0.825 | 82.28 | 6.205 | 0.431 |

### A.2.1 ANALYSIS OF ITERATIVE REFINEMENT

Subsequently, we evaluated the model's capacity for iterative improvement using the small-scale dataset. Specifically, we saved the model after each round of training. For each saved model, we generated 16 sequences for every backbone in the dataset, then calculates the values of Initial Guess, Evo. ppl, and AAR for each round's model on the small-scale test set.

According to Figure 5, MoMPNN[IG+ESM] exhibits greater stability than Weighted-score DPO across the three metrics (Initial Guess, Evo. ppl, and AAR) during the semi-online training process. Moreover, as training rounds incrementally increase, the Weighted-score DPO model tends to converge on a single metric. For instance, as shown in the Fig. 5, MoMPNN enables the joint optimization of the two objectives (reaching values of 0.589 and 6.154, respectively) while only causing a 1% decrease in AAR. In contrast, the optimization of Initial Guess for Weighted-score DPO exhibits fluctuations and starts to decline from the 6th round, with a more significant drop in

AAR. This indicates that MoMPNN can effectively optimize multiple objectives simultaneously while minimizing deviations from the base model.

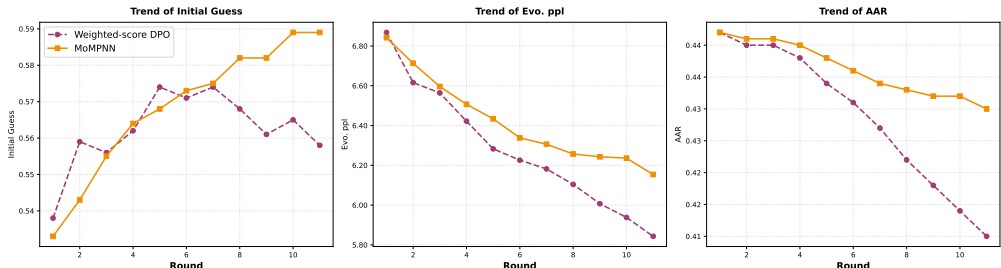

Figure 5: Analysis of MoMPNN [IG+ESM] and Weighted-score DPO. Initial Guess, Evo. ppl and recover rate changes across each round of iterative refinement.

# B  IMPLEMENT DETAILS

## B.1  MATHMATICAL DERIVATIONS

For the theoretical completeness, we provide some definitions, lemmas, theorems and proofs of all the formulas in the main text here (Zhou et al., 2023a; Rafailov et al., 2023).

Firstly, the multi-objective function is:

$$\arg\max_{\theta} \mathcal{L}(\pi_{\theta}) = \mathbb{E}_{x\sim\mathcal{D},y\sim\pi(y|x)}\left[\sum_{K} w_k r_k(x,y)\right] - \beta \mathbb{D}_{KL}(\pi_{\theta}(y|x)\|\pi_{\text{ref}}(y|x)). \tag{6}$$

Then, we further derive the above equation.

$$
\begin{aligned}
&\max_{\pi} \mathbb{E}_{x\sim\mathcal{D},y\sim\pi(y|x)}\left[\sum_{K} w_k r_k(x,y)\right] - \beta \mathbb{D}_{KL}(\pi(y|x)\|\pi_{\text{ref}}(y|x)) \\
&= \max_{\pi} \mathbb{E}_{x\sim\mathcal{D}}\mathbb{E}_{y\sim\pi(y|x)}\left[\sum_{K} w_k r_k(x,y)\right] - \beta \sum_{x\sim\mathcal{D},y\sim\pi(y|x)} \pi(y|x)\log\frac{\pi(y|x)}{\pi_{\text{ref}}(y|x)} \\
&= \min_{\pi} \mathbb{E}_{x\sim\mathcal{D}}\mathbb{E}_{y\sim\pi(y|x)}\left[\log\frac{\pi(y|x)}{\pi_{\text{ref}}(y|x)} - \frac{1}{\beta}\sum_{K} w_k r_k(x,y)\right] \\
&= \min_{\pi} \mathbb{E}_{x\sim\mathcal{D}}\mathbb{E}_{y\sim\pi(y|x)}\left[\log\frac{\pi(y|x)}{\frac{1}{Z(x)}\pi_{\text{ref}}(y|x)e^{\frac{1}{\beta}\sum_{K} w_k r_k(x,y)}} - \log Z(x)\right]
\end{aligned}
\tag{7}
$$

Among them,

$$Z(x) = \sum_{y} \pi_{\text{ref}}(y|x)e^{\frac{1}{\beta}\sum_{K} w_k r_k(x,y)} \tag{8}$$

so, we can define that:

$$\pi^*(y|x) = \frac{1}{Z(x)}\pi_{\text{ref}}(y|x)e^{\frac{1}{\beta}\sum_{K} w_k r_k(x,y)} \tag{9}$$

Noting that $Z(x)$ and $\pi$ are independent, Eq.7 is given as follows:

$$
\begin{aligned}
&\min_{\pi} \mathbb{E}_{x\sim\mathcal{D}}\mathbb{E}_{y\sim\pi(y|x)}\left[\log\frac{\pi(y|x)}{\frac{1}{Z(x)}\pi_{\text{ref}}(y|x)e^{\frac{1}{\beta}\sum_{k} w_k r_k(x,y)}} - \log Z(x)\right] \\
&= \min_{\pi} \mathbb{E}_{x\sim\mathcal{D}}\left[\mathbb{D}_{\text{KL}}(\pi(y|x)\|\pi_*(y|x)) - \log Z(x)\right]
\end{aligned}
\tag{10}
$$

So, we get the minimum when:

$$\pi(y|x) = \pi^*(y|x) = \frac{1}{Z(x)}\pi_{\text{ref}}(y|x)e^{\frac{1}{\beta}\sum_K w_k r_k(x,y)} \tag{11}$$

Then, we can get that:

$$\sum_K w_k r_k(x,y) = \beta \log \frac{\pi(y|x)}{\pi_{\text{ref}}(y|x)} + \beta \log \mathbf{Z}(x) \tag{12}$$

where k is the number of properties.

Based on Eq. 3 and maximum likelihood estimation, for the k-th property we can get the reward loss:

$$\mathcal{L}_R(r_k, \mathcal{D}_k) = -\mathbb{E}_{(x,y_w,y_l)\sim\mathcal{D}}\big[\log\sigma(r_k(x,y_w) - r_k(x,y_l))\big] \tag{13}$$

where $\sigma$ is the logistic function and $r_k(x,y)$ is the implicit reward model.

It can be seen from Eqs. 12 and 13 that for the k-th properties:

$$r_k(x,y) = \frac{1}{w_k}\left[\beta \log \frac{\pi_\theta(y|x)}{\pi_{\text{ref}}(y|x)} + \beta \log Z(x) - \sum_{\substack{K,\\k'\neq k}} w_{k'} r_{k'}(x,y)\right]$$

$$\mathcal{L}_R(\theta; r_k; \mathcal{D}_k) = -\mathbb{E}_{(x,y_w,y_l)\sim\mathcal{D}_k}\left[\log\sigma\Big(\frac{1}{w_k}\Big(\beta \log \frac{\pi_\theta(y_w\mid x)}{\pi_{\text{ref}}(y_w\mid x)} - \beta \log \frac{\pi_\theta(y_l\mid x)}{\pi_{\text{ref}}(y_l\mid x)}\right.$$
$$\left. - \sum_{\substack{K,\\k'\neq k}} w_{k'}\big(r_{k'}(x,y_w) - r_{k'}(x,y_l)\big)\Big)\Big)\right], \tag{14}$$

Since there are K properties, the final multi-objective training objective is given by:

$$\mathcal{L}_{\text{MO}}(\theta; r; \mathcal{D}) = -\sum_K w_k \mathbb{E}_{(x,y_w,y_l)\sim\mathcal{D}_k}\left[\log\sigma\Big(\frac{1}{w_k}\Big(\beta \log \frac{\pi_\theta(y_w\mid x)}{\pi_{\text{ref}}(y_w\mid x)} - \beta \log \frac{\pi_\theta(y_l\mid x)}{\pi_{\text{ref}}(y_l\mid x)}\right.$$
$$\left. - \sum_{\substack{K,\\k'\neq k}} w_{k'}\big(r_{k'}(x,y_w) - r_{k'}(x,y_l)\big)\Big)\Big)\right], \tag{15}$$

## B.2 HYPERPARAMETER SETTINGS

Unless otherwise stated, all training runs utilized the Adam optimizer ($\beta_1 = 0.9$, $\beta_2 = 0.98$, $\epsilon = 10^{-9}$) with a learning rate of 5e-6 for 20 rounds (600 training steps per rounds). Training was distributed across eight NVIDIA 4090 GPUs, with a total batch size of 64. The $\beta$ for the DPO loss is set to $0.5$. The weights for each objective are set as following: 0.6 for IG and TM, 0.4 for all other objectives.

## B.3 BASELINES IMPLEMENTATION

**ESM-IF.** We employed the test script provided in the ESM GitHub repository (`https://github.com/facebookresearch/esm/tree/main/examples/inverse_folding`), with the model esm_if1_gvp4_t16_142M_UR50. Aside from the parameters configured for our testing (detailed above to enable comparative evaluation), all remaining parameter settings followed the default configurations supplied in the repository. The default sampling temperature is set as 0.1, which is the same as ProteinMPNN.

**InstructPLM.** We utilized the test script provided in the GitHub repository (`https://github.com/Eikor/InstructPLM`), with all other parameters following the default settings specified

therein. It is noted the default temperature in experiments is default 0.8 and top p is 0.9. We also report the results when adopting the same configuration as ProteinMPNN for a fair comparison, i.e. temperature $T = 0.1$ without top p, which show similar performance.

**ProteinMPNN.** ProteinMPNN provides multiple models based on distinct noise levels. For a more comprehensive comparison, we adopted the default ProteinMPNN model with $0.2$ Å noise. We use the testing scripts of ProteinMPNN from the ProteinMPNN GitHub repository (`https://github.com/dauparas/ProteinMPNN`). All other settings are default.

**ProteinDPO.** We implemented the testing scripts for ProteinDPO from the GitHub repository (https://github.com/evo-design/protein-dpo). All other parameters followed the default settings specified, with the temperature set to 0.1.

**SolubleMPNN.** We implemented the testing scripts of ProteinMPNN from the ProteinMPNN GitHub repository using the checkpoint of SolubleMPNN (`https://github.com/dauparas/ProteinMPNN`). As the same as ProteinMPNN, we choose the default model with $0.2$ Å noise. All other settings are default.

**HyperMPNN.** We get the model weights for he different training settings (added backbone noise) from the HyperMPNN GitHub repository (`https://github.com/meilerlab/HyperMPNN`) and use the testing scripts of the original ProteinMPNN. Similar to ProteinMPNN, we choose the default model with $0.2$ Å noise.

**Guidance Method.** We implemented a predictor-free guidance method following the approach in (Nisonoff et al., 2024). The predictor-guided rates can alternatively obtained in terms of conditional $\mathbf{R}_t(x, \tilde{x}|y)$ and unconditional $\mathbf{R}_t(x, \tilde{x})$, rates for $x \neq \tilde{x}$ in the form

$$\mathbf{R}_t^{(\gamma)}(x, \tilde{x}|y) = \mathbf{R}_t(x, \tilde{x}|y)^\gamma \mathbf{R}_t(x, \tilde{x})^{1-\gamma}, \tag{16}$$

As shown in this equation, the guided rates $\mathbf{R}_t^{(\gamma)}(x, \tilde{x}|y)$ generalize both the conditional $[\mathbf{R}_t^{(\gamma=1)}(x, \tilde{x}|y) = \mathbf{R}_t(x, \tilde{x}|y)]$ and unconditional $[\mathbf{R}_t^{(\gamma=0)}(x, \tilde{x}|y) = \mathbf{R}_t(x, \tilde{x})]$ rates. In our experiments, we used the ProteinMPNN as the unconditional model and SolubleMPNN / HyperMPNN as the contional model. We set $\gamma = 0.5$, with all other settings consistent with used for ProteinMPNN. We used the reference code provided in the paper (`https://github.com/hnisonoff/discrete_guidance/`).

**Weighted-score DPO.** We directly train a DPO model by obtaining a final score from a weighted combination of the ratings provided by different preference annotators. This aggregated score was then used to construct training pairs, on which we applied the standard DPO training procedure. All hyperparameters, including the importance weight $w_r$, were kept identical to those used in MoMPNN.

### B.4 PREFERENCE PREDICTORS

**Structural Consistency.** We download the ESMFold model from `https://github.com/facebookresearch/esm`, and TMalign (`https://zhanggroup.org/TM-align/`) is used for calculating the TM score between predicted structure and the input. For AlphaFold Initial Guess, we use the implementation from `https://github.com/nrbennet/dl_binder_design`.

**Evolutionary Plausibility.** We download the esm2_t33_650M_UR50D model from `https://huggingface.co/facebook/esm2_t33_650M_UR50D`, and calculate the sequence pseudo perplexity following Kantroo et al. (2025).

**Solubility.** We use download the Protein-Sol predictor from `https://protein-sol.manchester.ac.uk/software`. The predictor outputs a score ranging $[0, 1]$ for each sequence.

**Thermostability.** We download the TemBERTure model from `https://github.com/ibmm-unibe-ch/TemBERTure` and use the `temBERTure_CLS` mode. The model outputs a Thermophilic score ranging $[0, 1]$ for each sequence.

## C    BENCHMARK DETAILS

### C.1    CATH 4.3 BENCHMARK

**Data.**    We download the CATH4.3 benchmark dataset from `https://github.com/A4Bio/ ProteinInvBench/releases/tag/dataset_release`. The dataset is provided as a JSONL file and we convert it to PDB files by extracting the sequences and backbone atom coordinates.

**Metrics.**    We evaluate the designed proteins using the following metrics:

- *RMSD* measures the average structural deviation between the designed and reference structures. The two structures are first aligned using the Kabsch algorithm, and the deviation is computed on $C\alpha$ atoms (for backbone). The designed structures are predicted by ESMFold.

- *TM score* quantifies the global structural similarity between the designed and reference structures. In our experiments, we used ESMFold to predict the 3D structure of the designed sequence, and then computed the TM-score against the reference crystal structure.

- *pLDDT* represents the average per-atom confidence score. The values are extracted from the B-factor column of the ESMFold output.

- *Evo ppl* evaluates the evolutionary plausibility of the designed amino acid sequence. Lower perplexity values indicate closer alignment with natural protein sequence patterns, thereby reducing risks of aggregation or misfolding (see Appendix B.4).

- *Sol* estimates the solubility of the designed sequence. The scores are predicted by Protein-Sol (see Appendix B.4).

- *Thermo* estimates the thermostability of the designed sequence. The scores are predicted by TemBERTure (see Appendix B.4).

- *AAR* measures the averaged amino acid recovery, i.e., the fraction of residues in the predicted sequence matching the original sequence:

$$\text{AAR} = \frac{1}{L} \sum_{i=1}^{L} \mathbf{1}(x_i = y_i), \tag{17}$$

  where $L$ is the sequence length, $x_i$ is the amino acid at position $i$ in the predicted sequence, and $y_i$ is the corresponding residue in the reference sequence.

The metrics are reported as the average value across all generated sequences for the test set.

### C.2    *De novo* DESIGN BENCHMARK

**Data.**    For the *de novo* design benchmark, we generated protein backbones with lengths ranging from 50 to 500 residues using RFDiffusion (Watson et al., 2023). Specifically, we generated four distinct backbones for each length within this range (i.e., 50, 51, 52, . . . , 500 residues). All backbone generations were performed using the default parameters of RFDiffusion. There are total 1,824 backbones as *de novo* design benchmark dataset. These generated backbones were then used as inputs for subsequent modeling and evaluation in our experiments.

**Metrics.**    We use the same set of evaluation metrics as the CATH4.3 benchmark except AAR.

### C.3    BINDER DESIGN BENCHMARK

**Data.**    To further evaluate the performance of our model, we additionally adapted several challenging target proteins as benchmarks to validate our model (Pacesa et al., 2024; Zambaldi et al., 2024). There were six distinct protein targets, and binders were designed for each of these targets. Supplementary Table 4 provides detailed information for each target protein, including its PDB ID, relevant chain details, and the length range of the designed binders. The hotspot residues specify the desired interaction interface on the target protein. Following previous works, we used RFDiffusion to generate 100 unique backbones for each binder, guided by the defined hotspot residues and binder length range.

Table 4: Input structures, hotspot settings and binder lengths for benchmarks.

| Design Target | Input PDB | Target chain and residue numbers | Target Hotspot | Binder length(for benchmarks) |
|---|---|---|---|---|
| PD-L1 | 5O45 | A18-132 | A56, A115, A123 | 50-120 |
| SC2RBD | 6M0J | E333-526 | E485, E489, E494, E500, E505 | 50-120 |
| BHRF1 | 2WH6 | A2-158 | A65, A74, A77, A82, A85, A93 | 80-120 |
| PD-1 | AF2 prediction | A32-146 | A64, A126, A129, A133 | 80-150 |
| CLN1-14 | AF2 prediction | A1-188 | A31, A46, A55, A152 | 80-175 |

**Metrics.** In binder design, we evaluate models by sequence success rate, backbone success rate, evolutionary perplexity, and solubility.

- *Inter-chain PAE* measures the predicted error in the relative alignment between the binder and target chains, with lower values indicating more precise and stable binding conformations.

- *Overall Cα RMSD* measures the structural deviation of the designed binder's backbone from a reference structure, with smaller values indicating greater conformational consistency and fold stability.

- *Binder pLDDT* measures the local confidence of each residue's spatial arrangement in the binder, where values above 80 typically correspond to experimentally validated, thermodynamically stable regions.

- *Evolutionary Perplexity* measures the evolutionary plausibility of the binder's amino acid sequence, with lower values indicating closer alignment with natural protein sequence patterns.

- *Solubility* measures the ability of the binder to dissolve in aqueous environments, an essential property for experimental manipulation and potential therapeutic applications.

The sequence success rate refers to the proportion of generated binders deemed successful, where a binder sequence is defined as successful if it meets three criteria: binder sequence pLDDT > 80, inter-chain PAE < 10, and overall $C\alpha$ RMSD < 2 Å. The backbone success rate refers to the proportion of successfully designed backbones among all generated binder backbones, where a binder backbone is considered successful if any one of its designed sequences meets the sequence success criteria. Together, these metrics confirm that the designed binder not only folds correctly but also maintains functional utility in practical scenarios.

## D   DISCUSSION ON DESIGN CHOICES

Our design choices were guided by computational limits, empirical observations, and the need to keep the multi-objective optimization stable. For rollout number and sampling temperature, early validation suggested that changes to these hyperparameters had only limited impact on final performance when the total number of training iterations was fixed in MoMPNN. This pattern allowed us to adopt a moderate configuration without extensive tuning. For property weights, we placed slightly more emphasis on the primary design objective than on auxiliary ones, since reducing the weight of the latter further slowed convergence and weakened results under the same training budget.

The adaptive margin used in the $L_{MO}$ follows directly from the multi-objective function, and it depends only on the assigned weights and the property scores of the paired sequences. With both the dataset and the weights fixed, the margin for each pair can be precomputed. But we also could create dynamic versions of the adaptive margin, for instance, by changing the weights over time or using signals from recent optimization behavior. The dynamic variants could potentially allow the optimization to follow the Pareto front more closely. But, it makes the model training unstable, and it also requires careful control over how different objectives influence one another. Consequently, we employs fixed weights and precomputed margins, while adaptive margin schemes are left for future work.

For preference-pair construction, we sorted rollouts and paired the best half against the worst half, using a delta threshold to remove uncertain pairs. Since all properties are estimated by predictive models, small differences often reflect noise rather than clear preferences. Filtering these cases makes the training data more reliable, but it also means the data only focuses on clear differences. More

sophisticated sampling strategies, such as hard-negative selection or uncertainty-aware sampling, could mitigate under sampling of the ambiguous region; however, implementing them requires careful consideration of prediction noise and uncertainty. These ideas remain interesting directions to pursue beyond the scope of the current work.

## E    DISCUSSION ON FUNCTIONAL PROTEIN SEQUENCE DESIGN

Recent explorations in protein sequence design have begun to extend beyond backbone-based specifications by introducing functional constraints that capture chemically relevant exterior features. Currently, advanced functional protein sequence design methodologies, such as SurfPro (Song et al., 2024), BC-Design (Tang et al.), and SurfDesign (Wu et al.), explicitly integrate critical chemical features of the exterior surface (e.g., hydrophobicity, charge activity) that govern interactions with the environment or ligands into the sequence design task. Our ProtAlign framework has the potential to further optimize these functional protein design models by not only handling explicit functional constraints but also guiding the design of sequences toward a broader spectrum of desirable physicochemical properties. Objectives such as binding affinity could be incorporated to enhance the designed protein's interaction capability with the target ligand; likewise, adding objectives like stability would support feasibility in biological or industrial environments. Optimization from these two perspectives may enable the models to design multi-property functional proteins that are both high-affinity and practically viable.

## F    LLM USAGE

LLMs were only used as a general-purpose tool for language editing and polishing in the preparation of this manuscript. Specifically, their role was limited to optimizing the expression fluency, refining academic terminology consistency, and adjusting sentence structures of the text; they did not participate in any aspect of this research, including the conception of research ideas, design of experimental protocols, collection or analysis of experimental data, or derivation of research conclusions. All content modifications made by LLMs have been thoroughly reviewed and verified by the authors to ensure accuracy, consistency with the original research intent, and compliance with academic norms. The authors take full responsibility for the final content of this manuscript.

