# OpenReview forum: "Property-Driven Protein Inverse Folding with Multi-Objective Preference Alignment"
_ICLR.cc/2026/Conference — ICLR 2026 Poster_

### Official Review · Reviewer_qpSG · 2025-10-27

**Soundness:** 2
**Presentation:** 3
**Contribution:** 2
**Rating:** 4
**Confidence:** 3

**Summary:**

The paper addresses the challenge that protein inverse folding models must balance designability (recovering a backbone) with developability properties (e.g., solubility, thermostability). The authors propose ProtAlign, a multi-objective preference alignment framework that fine-tunes pretrained models using a semi-online Direct Preference Optimization (DPO) strategy. The method uses a flexible preference margin to mitigate conflicts between competing objectives and constructs preference pairs using in silico property predictors. Applying this to ProteinMPNN yields MoMPNN. Experiments on CATH 4.3 crystal structures , de novo backbones , and binder design scenarios show that MoMPNN enhances developability properties without compromising structural fidelity compared to baselines.

**Strengths:**

This method improves developability metrics using a preference alignment framework , which does not require additional specific, curated datasets of experimentally-validated proteins.

The authors evaluate MoMPNN on a strong set of tasks beyond standard sequence recovery. This includes redesigning CATH 4.3 crystal structures, designing sequences for de novo generated backbones, and a practical de novo binder design scenario. This rigorous evaluation demonstrates the method's utility in realistic design workflows where other baselines show performance degradation.

**Weaknesses:**

It would be better to report the metrics on ground truth sequences, as these metrics are based on prediction models as approximations.

Full names of abbr.’s in tables are missing in the captions.

The temperatures used in inference of different baselines are not identical, resulting in potentially unfair comparison. A fair comparison would be either the greedy strategy (without temperature), or comparing the best point on the temperature-performance curves between different methods; or at least report the results under one identical temperature.

is it a typo in eq 4? k appears in the formula of m, which seems irrelevant to k.

Explanation for the relationship between L and L_MO is needed.

**Questions:**

Why is the AAR of ProteinMPNN on CATH 4.3 test 0.39, which seems lower than most of the reproduction of this model, e.g., 0.44 on CATH 4.3 was reported in ProteinInvBench? If this AAR is not correct, does it indicate a significant compromise of AAR?

RL-based preference methods like ProteinDPO for inverse folding are discussed in the related work section. Why are they compared as baselines? They are supposed to be the most related baselines.

Regarding the semi-online training strategy, is the preference dataset $\mathcal{D}_k$ at iteration t cumulative (containing all rollouts from iterations $1 \dots t$), or is it replaced entirely by the new rollouts?

The paper provides a compelling comparison against a "Weighted-score DPO" baseline in Appendix A.2, showing MoMPNN is more stable. Can the authors provide more intuition on why the flexible margin (Eq. 4 ) achieves better and more stable multi-objective optimization compared to directly optimizing on a weighted sum of preference scores?

The model is trained on protein monomers but evaluated on a de novo binder design task, which involves protein complexes. Did the authors observe any specific failure modes or performance issues at the binder-target interface, given that the model was not explicitly trained for complex-specific properties?

---

> ### Author Response · Authors · 2025-11-19
> **Official Comment by Authors (1/3)**
>
> Thank you for your careful review and valuable suggestions. Below we respond to each point in detail.
>
> ### **1. On Experimental Results, Baselines, and Additional Evaluations (W1, W3, Q1, Q2)**
>
> In response to your suggestions, we have included the additional results summarized in the following table. The new results will be updated to the experiment section of our paper. These updates correspond to the concerns raised in **W1**, **W3**, and **Q2**, and we elaborate on each of them in the sections below.
>
> |                      | RMSD  | TM    | plddt | PPL   | sol   | thermo | aar   |
> | -------------------- | ----- | ----- | ----- | ----- | ----- | ------ | ----- |
> | **Cath 4.3 Test**    |       |       |       |       |       |        |       |
> | ground truth         | 3.97  | 0.761 | 80.8  | 5.80  | 0.620 | 0.246  | 1.000 |
> | ProteinDPO           | 5.49  | 0.667 | 72.0  | 10.50 | 0.629 | 0.357  | 0.388 |
> | InstructPLM(tem=0.8) | 6.81  | 0.628 | 73.4  | 7.97  | 0.653 | 0.396  | 0.574 |
> | InstructPLM(tem=0.1) | 6.96  | 0.632 | 74.4  | 7.31  | 0.657 | 0.455  | 0.584 |
> | MoMPNN[Sol+IG+EP]    | 4.61  | 0.731 | 79.3  | 5.99  | 0.856 | 0.789  | 0.384 |
> | MoMPNN[Thermo+IG+EP] | 4.37  | 0.737 | 78.5  | 5.97  | 0.723 | 0.947  | 0.385 |
> | **De novo Task**     |       |       |       |       |       |        |       |
> | ProteinDPO           | 16.77 | 0.296 | 43.5  | 14.70 | 0.596 | 0.145  | NA    |
> | InstructPLM(tem=0.8) | 22.44 | 0.134 | 32.6  | 3.33  | 0.539 | 0.278  | NA    |
> | InstructPLM(tem=0.1) | 22.58 | 0.132 | 33.5  | 2.73  | 0.538 | 0.367  | NA    |
> | MoMPNN[Sol+IG+EP]    | 6.17  | 0.751 | 72.0  | 7.34  | 0.843 | 0.993  | NA    |
> | MoMPNN[Thermo+IG+EP] | 6.20  | 0.748 | 71.2  | 7.44  | 0.723 | 0.998  | NA    |
>
> **(a) Ground-truth metrics (W1)**
>
> We agree that reporting ground-truth metrics provides a clearer reference, and we have added these results on the CATH 4.3 test set. It is also worth noting that some ground-truth sequences in CATH are not soluble, and most of them are not thermally stable, so the solubility and thermostability of ground-truth sequences are low.
>
> **(b) Inference temperature consistency (W3)**
>
> - ESM-IF is evaluated with its standard inference temperature of 0.1, consistent with ProteinMPNN. We've added this to the appendix.
> - For InstructPLM, we added results at temperature 0.1, which show similar performance and do not affect any of our conclusions.
>
> **(c) RL-based preference learning baselines (Q2)**
>
> We appreciate your pointing out that RL-based DPO methods are closely related. To the best of our knowledge, ProteinDPO is the only work in this category with publicly available code. Although it does have sequence-generation capability, its primary focus is stability prediction for mutant variants of a given protein, rather than general inverse folding, so we did not include it in our main experiments of initial submission.
>
> For completeness, we have added its results to the updated table. As shown there, its performance is slightly worse than ESM-IF, the model it is finetuned from, and does not affect any of our conclusions.

---

> ### Author Response · Authors · 2025-11-19
> **Official Comment by Authors (2/3)**
>
> ### **2. On Generalization to De Novo Binder Design (Q5)**
>
> We did not observe any notable failure modes at the binder–target interface, although the results could likely be further improved by directly optimizing interface quality with multi-chain data and an appropriate metric. It is likely that the DPO's regularization toward the reference model helps the model learn a more general pattern for improving generated sequences, which may explain why it generalizes well even without complex-specific training, and the reference model ProteinMPNN is trained on multi-chain datasets.
>
> To further analyze the binder-target interface, we utilized AlphaFold-Multimer to calculate the mean ipAE and ipTM, which are used to assess the quality of the predicted binder-target interface within the complex structure. Lower ipAE or higher ipTM values are strongly correlated with docking quality and are therefore widely used as key metrics for evaluating the success of de novo binder design [1,2].
>
> For each binder, we generated 800 sequences. The summary statistics for the ipAE and ipTM across these 800 sequences are presented in the table below.
>
> The results indicate that the ipAE and ipTM of the protein sequences designed by MoMPNN model is comparable to that of SolubleMPNN and marginally superior to ProteinMPNN. Crucially, the maximum ipAE and minmum ipTM for MoMPNN is consistently and marginally better than that of ProteinMPNN across all five binders, and remains comparable to SolubleMPNN. This observation confirms that even with training conducted exclusively on monomer structures, the model's capability to predict or optimize complex structures was not compromised.
>
> | ipAE Avg/Mid/Max (lower better) | PD-1                | BBF-14              | BHRF1             | SC2RBD              | PDL1                |
> | -------------------------------- | ------------------- | ------------------- | ----------------- | ------------------- | ------------------- |
> | ProteinMPNN                      | 18.31/ 20.53/ 29.13 | 22.81/ 24.79/ 29.38 | 8.65/ 7.73/ 27.57 | 21.90/ 24.29/ 28.47 | 14.34/ 14.38/ 26.45 |
> | SolubleMPNN                      | 16.39/ 16.90/ 29.59 | 22.06/ 24.35/ 29.74 | 8.12/ 7.17/ 19.90 | 20.36/ 23.56/ 28.91 | 13.90/ 14.09/ 26.37 |
> | MoMPNN                           | 16.51/ 16.20/ 29.30 | 22.12/ 24.53/ 29.09 | 8.23/ 7.36/ 23.86 | 21.06/ 24.14/ 27.77 | 12.79/ 11.89/ 26.44 |
>
> | ipTM Avg/Mid/Min (higher better) | PD-1             | BBF-14           | BHRF1             | SC2RBD           | PDL1             |
> | ------------------------------- | ---------------- | ---------------- | ----------------- | ---------------- | ---------------- |
> | ProteinMPNN                     | 0.33/ 0.21/ 0.05 | 0.21/ 0.13/ 0.06 | 0.80/ 0.82/ 0.08  | 0.24/ 0.15/ 0.08 | 0.42/ 0.36/ 0.07 |
> | SolubleMPNN                     | 0.39/ 0.34/ 0.05 | 0.23/ 0.14/ 0.06 | 0.82/ 0.83 / 0.47 | 0.29/ 0.17/ 0.07 | 0.43/ 0.37/ 0.07 |
> | MoMPNN                          | 0.39/ 0.35/ 0.05 | 0.23/ 0.14/ 0.06 | 0.81/ 0.83 / 0.16 | 0.25/ 0.16/ 0.08 | 0.47/ 0.46/ 0.07 |

---

> > ### Author Response · Authors · 2025-11-19
> > **Official Comment by Authors (3/3)**
> >
> > ### **3.AAR discrepancy for the original ProteinMPNN (Q1)**
> >
> > The AAR discrepancy between our reported results and ProteinInvBench [3] arises from the use of different model checkpoints. Our AAR of 0.39 on CATH 4.3 is obtained from the default ProteinMPNN checkpoint trained with 0.20 Å noise (v_48_020.pt in the official repository). In contrast, the 0.44 reported in Table 1 of ProteinInvBench comes from a re-trained ProteinMPNN on CATH 4.3 without any added noise.
> >
> > The noise-added default version of ProteinMPNN is known to provide more robust out-of-distribution behavior in real-world settings, and it has become a commonly adopted standard in industry.
> >
> > The Table 4 of ProteinInvBench also reports results for models trained with different noise magnitudes. Notably, the ProteinMPNN model re-trained with 0.20 Å noise achieves an AAR of 0.35, which is lower than the 0.39 we obtain using the official pre-trained weights.
> >
> > We hope this helps clarify that MoMPNN’s improved developability does not stem from a compromise in AAR.
> >
> > ### **4. On the Semi-Online Training Dataset (Q3)**
> >
> > The preference dataset is not cumulative. It will be replaced entirely by the new rollouts. At iteration t, all the data used for model training was generated by the model obtained at iteration t−1. We also modified Algorithm 1 to make it clearer.
> >
> > ### **5. On Flexible Margin vs. Weighted-Sum DPO (Q4)**
> >
> > Weighted sum of preference scores works by creating a single objective, and it optimizes for the best sum score. This often leads to final solutions clustered around a single point, failing to find the true, diverse set of optimal compromises, as optimal compromises may be the non-linear combination of objectives.
> >
> > For two-objective optimization, Weighted-Sum DPO may continue to optimize Objective A, even if its results are already excellent, while the results for Objective B remain poor. This occurs because the method optimizes the sum of objectives A and B, and thus does not account for the non-linear relationship or trade-off between A and B. In contrast, the multiple objectives in MoMPNN are inherently non-linear. The model calculates the loss for objectives A and B separately, and the loss function itself can capture the potential non-linear relationship between A and B (the adaptive margin in Eq 4). Therefore, it can achieve simultaneous optimization of both Objective A and Objective B.
> >
> > ### **6. On Table Captions and Notation Consistency (W2, W4)**
> >
> > We have added full names of all abbreviations in table captions and corrected the noted typographical issues. $L_{MO}$ in Equation 4 and $L$ in Algorithm 1 convey the same meaning, and we have now corrected this error in the original text.
> >
> > Thank you again for your time and valuable insights! We look forward to hearing from you.
> >
> > [1] Genz, L. R., Nair, S., Nagar, N., & Topf, M. (2025). Assessing scoring metrics for AlphaFold2 and AlphaFold3 protein complex predictions. *Protein Science*, *34*(11), e70327.
> >
> > [2] Zambaldi, V., La, D., Chu, A. E., Patani, H., Danson, A. E., Kwan, T. O., ... & Wang, J. (2024). De novo design of high-affinity protein binders with AlphaProteo. arXiv preprint arXiv:2409.08022.
> >
> > [3] Gao, Z., Tan, C., Zhang, Y., Chen, X., Wu, L., & Li, S. Z. (2023). Proteininvbench: Benchmarking protein inverse folding on diverse tasks, models, and metrics. *Advances in Neural Information Processing Systems*,*36*, 68207-68220.

---

> > > ### Comment · Reviewer_qpSG · 2025-11-19
> > >
> > > Thank you. The response makes sense to me, addressed issues and answered my questions. Considering that the method is function-oriented, thus has the potential for broader impact on protein sequence design models, rather than just inverse folding, i suggest adding introduction to the functional protein sequence design approaches in the related work section, and discuss the potential of your work in being applied to such models in the conclusion section or an additional discussion section. Examples are:
> > >
> > > - SurfPro: Functional Protein Design Based on Continuous Surface
> > > - BC-Design: A Biochemistry-Aware Framework for Highly Accurate Inverse Protein Folding
> > > - SurfDesign: Effective Protein Design on Molecular Surfaces

---

> > > > ### Author Response · Authors · 2025-11-20
> > > >
> > > > It’s great to hear that our earlier reply helped address your concern and thank you for the suggestion.
> > > >
> > > > We have updated the paper accordingly. In the Related Work section, we now include a concise discussion of functional protein sequence design as an extension of traditional inverse folding, highlighting how recent approaches such as SurfPro, BC-Design, and SurfDesign incorporate biochemical surface features into the design process. We also added a short paragraph in the Appendix describing how our framework can further improve these models by supporting optimization over broader developability objectives.
> > > >
> > > > If anything else feels unclear or could be improved, please feel free to let us know!

---

> > > > > ### Comment · Reviewer_qpSG · 2025-11-23
> > > > >
> > > > > I have no further comments and I chose to raise the rating score.

---

### Official Review · Reviewer_4A1d · 2025-10-27

**Soundness:** 3
**Presentation:** 3
**Contribution:** 3
**Rating:** 6
**Confidence:** 4

**Summary:**

The paper proposes ProtAlign, a multi-objective preference-alignment framework for protein inverse folding that optimizes developability properties without compromising designability. The method uses a semi-online DPO loop: generate rollouts at higher temperature, score them with property predictors, construct pairwise preferences per property, then train offline with an adaptive preference margin to reconcile conflicts among objectives. Instantiated on ProteinMPNN as MoMPNN, the approach is evaluated on CATH 4.3, de novo backbones from RFDiffusion, and realistic de novo binders; results show developability gains while maintaining or improving structural consistency relative to strong baselines.

**Strengths:**

- Method is simple and general: multi-objective DPO with an adaptive preference margin to mitigate conflicts across properties; the training pipeline evenly samples pairwise entries across properties and alternates rollout and training for efficiency.

- Practical semi-online training decouples rollout/evaluation from optimization, enabling batch computation and easier deployment while retaining online exploration benefits.

- Evaluations are broad and application-relevant: crystal redesign, de novo backbones, and realistic binder design; the study systematically integrates developability metrics into inverse-folding evaluation beyond amino acid recovery.

- The presentation style is good, with nice-looking figures and easy-to-follow-up narration styles.

**Weaknesses:**

- Limited ablations on multi-objective weights and margin settings. It might be helpful to quantify how weights, temperature, and margin thresholds shape the Pareto front and to provide transferable default configurations as the paper heavily relies on it.
- The adaptive preference margin m(yw,yl) is precomputed from auxiliary property deltas and then kept fixed during training. This is simple and fast, but it cannot react if the policy distribution drifts, predictors recalibrate, or property trade-offs evolve; the “right” margin may change as the frontier moves.
- Pair construction may over-represent “easy wins” and under-sample ambiguous regions. Preference pairs are formed by sorting rollouts and pairing top-half vs. bottom-half, with a delta threshold to drop uncertain pairs. While this stabilizes supervision, it can bias learning away from the decision boundary where the frontier is decided. Active pair mining (hard-negative selection) or uncertainty-aware sampling could help learn more from the ambiguous region and reduce label imbalance across properties.

**Questions:**

- Can the weights across properties and the adaptive margin be tuned online using objective-improvement rates to more reliably approach a Pareto front across backbones and lengths?

- What is the effect of the number of rollouts and sampling temperature on the stability of training and final metrics in the semi-online loop, given that the paper uses a higher temperature for exploration but evaluates at a lower temperature for ProteinMPNN-family models?

---

> ### Author Response · Authors · 2025-11-19
>
> We sincerely thank you for your thoughtful feedback and constructive suggestions. Below, we provide detailed responses to each point. The discussions on ablation studies, design choices, and future research directions will be added to the final version of our paper.
>
> ### **1. On ablations of multi-objective weights, rollout number, and sampling temperature (W1, Q2)**
>
> Because each round of online training requires fresh rollouts, our computational budget limits the scale of ablations we can perform. During development, we carried out a set of validation-set experiments to understand how the training dynamics behave under different hyperparameter settings.
>
> **(i) Rollout number & sampling temperature.**
>
> In our preliminary studies, these two factors had only modest influence on final performance when the total number of training iterations was kept fixed, suggesting that our method is not particularly sensitive to these hyperparameter choices. For example, results on the validation set for our solubility variant Sol+TM shows that:
>
> - *Rollout number*
>   - 8 rollouts (default): solubility 0.882, tmscore 0.826, esm ppl 6.61
>   - 16 rollouts: solubility 0.879, tmscore 0.821, esm ppl 6.60
> - *Sampling temperature*
>   - temperature = 1.0 (default): results above
>   - temperature = 0.1: solubility 0.871, tmscore 0.824, esm ppl 6.75
>
> Taking both computation and diversity into account, we chose the current configuration and did not tune them carefully for better results.
>
> **(ii) Property weights.**
>
> Weights across properties affect convergence speed. We tested reducing the developability weight from 0.4 to 0.2, and the model converged much slower and failed to match the original performance within 20 training iterations. When choosing weights, we set developability slightly lower (0.4) than designability (0.6) to keep designability as the primary objective in the inverse folding task, and this setup produced a good balance between the optimization goals.
>
> ### **2. On dynamic tuning of weights and adaptive margins (W2, Q1)**
>
> We obtain the DPO loss with an adaptive margin (Eq 4) from the weighted multi-objective optimization (detailed in Appendix B.1). According to the Eq 4, the adaptive preference margin $m_k(y_w,y_l)$ is solely a function of the weights and the sequences' property scores. Consequently, within a single optimization round, this margin is constant and can be precomputed, provided the multi-objective dataset and weights are fixed.
>
> The margin could also be made more "dynamic": for example, by adapting the weights or by adding a small extra term based on the improvement trends from recent iterations. This would in turn produce dynamically changing margins and may lead to a better training result. Such an approach may help the model pay more attention to objectives that lag behind, potentially allowing the optimization to follow the Pareto front more closely. However, it also introduces new challenges: dynamic weights may destabilize training, and any adaptation must still respect the relative importance of different objectives. This would require a carefully designed weight-adjustment scheme.
>
> In this work, we therefore adopt fixed weights and precomputed margins so that the optimization dynamics remain predictable under our limited computational budget. While this strategy already yields satisfactory performance, we agree that more adaptive approaches are promising and plan to investigate them in future work. We expect that a dynamic margin design could offer additional benefits in more challenging settings.
>
> ### **3. On pair construction and potential bias toward “easy wins” (W3)**
>
> Our current pairing strategy was chosen mainly because all property scores come from predictive models and therefore carry noise. When two sequences differ only slightly, we cannot confidently say one is better, so including such pairs often introduces unreliable supervision, which is consistent with current RLHF workflows [1]. Filtering them for high confidence pairs helps keep the learning signal cleaner.
>
> We agree this strategy may under-sample ambiguous regions and bias the supervision toward clearer wins. Techniques such as hard-negative mining and uncertainty-aware sampling could reduce this imbalance. Because the underlying scores contain prediction noise, these approaches require careful design and may call for more sophisticated balance between noise and uncertainty similar to [2,3]. We view them as valuable directions and plan to explore the use of confidence or uncertainty estimates in future work.
>
> Thank you again for your time and valuable insights! We look forward to hearing from you.
>
> [1] Zhang, Chuheng, et al. "Policy filtration for rlhf to mitigate noise in reward models.". 2024.
>
> [2] Oh, Minhyeon, et al. "Comparison-based Active Preference Learning for Multi-dimensional Personalization.". 2025.
>
> [3] Holladay, Rachel, et al. "Active comparison based learning incorporating user uncertainty and noise.". 2016.

---

> > ### Comment · Reviewer_4A1d · 2025-11-21
> > **Nice rebuttal**
> >
> > Thanks for the detailed rebuttal. My major concerns have been resolved, I personally would lean to accept this paper.

---

### Official Review · Reviewer_NtsP · 2025-10-28

**Soundness:** 3
**Presentation:** 3
**Contribution:** 3
**Rating:** 8
**Confidence:** 3

**Summary:**

This paper applies multi-objective preference optimization to protein inverse folding, using semi-online DPO with adaptive margins to balance structural accuracy against properties like solubility and thermostability. The resulting model, MoMPNN, beats existing baselines  across several benchmarks. The approach is solid but not particularly novel—it's essentially transplanting techniques from LLM alignment into protein design. That said, the execution is strong: the experiments are thorough, the amino acid distribution analysis shows the model learns sensible patterns, and the framework appears general enough to extend to other properties. The comprehensive evaluation is strong.

**Strengths:**

See summary

**Weaknesses:**

See summary

**Questions:**

No questions.

---

> ### Author Response · Authors · 2025-11-19
>
> Thank you sincerely for the kind and encouraging review. We are very grateful for the attention you gave the work, and your positive assessment really means a lot to us. We appreciate the recognition of the experimental thoroughness and the strength of the evaluation.

---

### Author Response · Authors · 2025-11-28
**General Response**

We greatly appreciate the reviewers' time and constructive feedback. We have addressed their specific comments point-by-point in the responses, and all reviewers have confirmed that their concerns have been satisfactorily resolved. Below, we provide a summary of the main revisions in the updated paper. All revised portions are highlighted in red for easy review.

- **Discussion on design choices  (Reviewer 4A1d)**: We have added the relevant discussion and rationale on the pairwise data construction and hyperparameter tuning in **Appendix D**.

- **Additional results and clarifications (Reviewer qpSG)**: We have added the results for ProteinDPO, InstructPLM (temperature=0.1), and the ground-truth data of the CATH 4.3 test set as additional baselines in **Section 5.2 and 5.3**. These new results do not affect any of our conclusions and help present a more complete picture of the experiments. We have also explicitly clarified these temperature settings in **Section B.3**.

- **Applicability on functional protein sequence design models (Reviewer qpSG)**: We have included content regarding this in **Section 2** and added a discussion in **Appendix E**.

We want to thank all reviewers again for their careful and constructive comments, and we believe these revisions have improved the clarity and strengthened our work.

---

### Meta-Review · Area_Chair_nPxd · 2026-01-06

**Summary:**

This paper proposes ProtAlign, a multi-objective preference alignment framework that fine-tunes pretrained inverse folding models to improve developability. Using a semi-online DPO approach and preference pairs constructed from in silico property predictors, the method avoids heavy manual tuning and target-specific retraining. When applied to ProteinMPNN, the resulting model improves developability across a range of design tasks without sacrificing structural accuracy.

It can be seen from the reviews and author-reviewer discussion that the reviewers are convinced to increase the score.

**Reviewer Concerns:**

Reviewer 4A1d's concern is mostly about limited ablations, and I believe the authors have addressed it during the rebuttal.

Reviewer qpSG's concern is mostly about lacking details in the experiments, and I believe the authors have added a lot more details during the rebuttal and paper revision.

**Reviewer Scores:**

Reviewer NtsP gave 8, while their review is short and not informative. I don't consider it when I make recommendation.

Reviewer Reviewer 4A1d gave 6, they said "My major concerns have been resolved, I personally would lean to accept this paper."

Reviewer qpSG gave 4, but they said "I have no further comments and I chose to raise the rating score." So I predict they will increase the score from 4 to 6.

---

### Decision · Program_Chairs · 2026-01-26

Accept (Poster)